# Effects of Social Support on Professional Identity of Secondary Vocational Students Major in Preschool Nursery Teacher Program: A Chain Mediating Model of Psychological Adjustment and School Belonging

**Yingxin Chen [1], Huihua He [1,\*] and Yan Yang [2]** 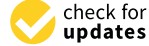

1 Shanghai Institute of Early Childhood Education, Shanghai Normal University, Shanghai 201418, China; 1000511618@smail.shnu.edu.cn
2 Tianhua College, Shanghai Normal University, Shanghai 201807, China; yy2042@sthu.edu.cn
\* Correspondence: hehuihua@shnu.edu.cn

**Abstract:** Background: In the context of the high turnover rate of preschool nursery teachers, the preschool nursery teacher program in secondary vocational schools has been an important channel for sending qualified nursery teachers to early education institutions, and fostering students' professional identity, which is key to their future career construction and development; therefore, this study aims to investigate the mechanism of social support on professional identity, with a chain mediating effect of psychological adjustment and school belonging. Methods: 377 third-year students from secondary vocational schools majoring in nursery and preschool teaching were surveyed with the Social Support Scale, Professional Identity Scale, School Belonging Scale, Well-being Scale, Discrimination Perception Scale and Self-Identity Scale. Results: (1) Correlation analysis showed that social support, professional identity, school belonging, and psychological adjustment (well-being, self-identity) were significantly and positively correlated with each other. A significant negative correlation was found between discrimination perception and other variables. (2) School belonging played a partially mediating role between social support and professional identity, and psychological adjustment and school belonging showed chain mediating effects between social support and professional identity. Conclusions: Social support not only directly influences professional identity, but also indirectly affects professional identity through the chain mediating effect of psychological adjustment and school belonging.

**Keywords:** professional identity; social support; psychological adjustment; school belonging; preschool nursery teacher

## 1. Introduction

Vocational education in Mainland China is an indispensable component of the educational system in the process of Chinese-style modernization, which is responsible for cultivating diversified professionals, inheriting handicrafts, and creating employment opportunities. However, secondary vocational school students are often seen as losers under the Chinese examination system, and although some researchers argue that it is stereotypical and discriminatory to view secondary vocational school students as a group with a low professional learning ability [1], it is also a real problem that secondary vocational school students have a lower sense of learning self-efficacy, and lack confidence in reaching their learning goals and producing positive expected learning outcomes compared to general education students [2]. The preschool nursery teacher major in secondary vocational schools is a type of significant teacher education program, which could be a critical assurance for implementing the unique integration of nursery care and early education in preschools domestically. Nonetheless, previous research has indicated that the social status, income, and work intensity of preschool nursery teachers are underestimated [3,4].

Learning motivation is often linked to career development status, students being trapped in a poor employment situation, students being young and lacking certain learning goals and motivation, students having some confidence in the learning preschool program but not high [5], and secondary vocational school students in the preschool field having a certain degree of academic burnout in learning the profession and not being able to obtain a high sense of achievement from their professional learning [6]. These realistic problems will make it difficult for students to form a high professional identity. Therefore, the professional identity of the students majoring in the preschool nursery teacher program in secondary vocational schools deserves extensive attention.

Professional identity is an emotional acceptance and recognition that is based on learners' perceptions of the profession they are studying, accompanied by positive external behaviors and internal matching. It is an integrated psychological process composed of emotions, attitudes, perceptions, and behaviors that serves as the foundation for the development of professional identity; it is also an important psychological representation of internal motivation for career development [7]. Professional identity and vocational identity are different. First, "vocational" studies emphasize occupations with the function of earning a living, while "professional" studies emphasize professional occupations learned via training and practice, such as preschool nursery teachers [8]. Secondly, secondary vocational school students have only been exposed to and trained in professional skills, but have not yet been involved in real vocational work; indeed, early adolescents are at the stage of forming a vocational identity and exploring careers, but not of committing to a career [9]. Finally, intervening earlier in students' professional identities is more beneficial to the development of future preschool nurse teachers' vocational identity at its foundation. Therefore, the term "professional identity" was used in this study. Professional identity influences not only the academic learning and career identification of secondary vocational school students [10], but also their job satisfaction and long-term desire to be a teacher [11,12]. Furthermore, secondary vocational school students are considered underprivileged learners by their peers and have long been in the predicament of not being recognized by society, having extremely limited professional development, and having little self-perceived social support [13].

Social support includes emotional, informational, evaluative, and instrumental support that individuals perceive to receive from others in their social network, such as parents, teachers, and peers [14]. Social support is proven to have a significant impact on the developmental outcomes of children and adolescents, which is a significant predictor of job preparation [15]. Adolescent students in secondary vocational schools have a specific need for multidimensional social support in order to reinforce their self-awareness and employment preparation. However, the mechanism underlying the relationship between social support and professional identity remains unclear. According to the perspective of career construction, social support will promote adaptive behaviors and outcomes by increasing individuals' adaptive abilities, while those with stronger adaptive abilities will acquire a professional identity through job exploration and career commitment [16]. Regarding the theory of career construction, revealing the influence mechanism of social support on professional identity is beneficial to secondary vocational school students' social adaptation and career construction, laying the psychological acceptance and commitment foundations for their career development.

### 1.1. Social Support and Professional Identity

Due to the absence of social experience, adolescents are often not thoroughly aware of their major and future careers. The primary sources of students' professional identity are interpretations or descriptions from family members, teachers, and peers. Prior studies have demonstrated that social support could positively predict students' professional identity [17]. Particularly, financial and emotional support from parents, teachers, classmates, and friends can increase students' sense of security and self-confidence, enable them to have a more positive mindset towards their majors, and motivate them to have

more positive cognition and emotion towards their majors. Moreover, social support also alleviates students' academic stress, adjusts negative emotions toward their majors, and promotes students' learning of professional knowledge [18,19]. Therefore, the current study hypothesizes H1 as follows:

**H1.** *Social support positively predicts the professional identity of secondary vocational students majoring in preschool nursery teacher programs.*

### 1.2. Psychological Adjustment as a Mediator

Psychological adjustment refers to the condition of psychological and behavioral responses to changes in the environment and in the body and mind, and is a dynamic response made by the individual to adapt to changes so that he or she can adjust himself or herself to maintain balance with the environment [20]. It can be commonly explained by positive or negative psychological outcomes, such as well-being, life satisfaction, identity integration, depression, and anxiety [21]. Previous research has taken well-being, self-esteem, depression, and anxiety as indicators of psychological adjustment [22,23], and as perceived discrimination is the stressor of mental health issues such as depression and anxiety, it is potentially a vital predictor of psychological adjustment [24]. Therefore, the present study employs well-being, self-identity, and discrimination perceptions as indicators of psychological adjustment.

The literature has indicated that social support could facilitate psychological adjustment. For example, adolescents who receive more social support tend to have greater well-being and self-identity [25,26]. Additionally, parental and peer support could reduce negative outcomes in adolescents, such as loneliness and depression caused by individual or group discrimination [27]. Furthermore, psychological adjustment may affect students' professional identity, that is, greater happiness, a positive self-identity, and the perception of less discrimination create the sense of a stronger professional identity [17,26,28,29]. As students' psychological well-being improves, their professional anxiety reduces, resulting in a stronger sense of professional identity [30]. Therefore, this paper hypothesizes H2 as follows:

**H2.** *Psychological adjustment plays a mediating role between social support and professional identity.*

### 1.3. School Belonging as a Mediator

School belonging is the degree to which students feel respected, included, and supported in the school environment [31]. Self-determination theory suggests that when students are satisfied with the support of others, their autonomy and sense of belonging is satisfied, fostering greater intrinsic motivation; this generates greater interest in school and learning, as well as positive autonomous behaviors [32]. As a result, students can strengthen their interest in school and influence their professional identity while receiving more social support. It has been suggested that supportive behaviors in parents, teachers and classmates predict students' attitudes and sense of belonging in school [14,33]. Specifically, in supportive and inclusive learning environments, students experience high-quality interpersonal interactions and relationships that lead to a positive sense of school belonging, while in unsupported or even threatening environments, students tend to feel marginalized and excluded, leading to an absence of school belonging [34]. Although little empirical research has discussed the predictive role of school belonging in fostering professional identity, positive relationships between school belonging, academic engagement and outcomes, professional development, and career motivation and expectations have been illustrated [35,36]. School affiliation improves the professional commitment of student teachers, thereby increasing an individual's recognition of their professional goals and values [37]. These findings indirectly show that students' professional identity is able to be

enhanced by their sense of belonging in school. Therefore, the current study hypothesizes H3 as follows.

**H3.** *School belonging plays a mediating role between social support and professional identity.*

*1.4. The Chain Mediation Effects of Psychological Adjustment and School Belonging*

Previous studies have verified that social support positively affects professional identity and the results are relatively consistent, but its internal mechanism of action is not clear. Some studies have also found that social support affects psychological adjustment and school belonging, and psychological adjustment and school belonging affect professional identity; therefore, the chain mediation effect of psychological adjustment and school belonging is proposed to investigate the mechanism of action of social support on professional identity. Although numerous studies have used social support as a moderator to mitigate the problems of student psychological adjustment and school belonging, few studies have incorporated both variables of psychological adjustment and school belonging into the mechanisms underlying the relationship between social support and professional identity. Moreover, a positive self-identity, a higher sense of well-being, and the perception of less discrimination are important factors that affect school belonging [22,23,38,39]. Empirical studies have shown that positive self-perception and a greater sense of self-identity positively predict school belonging [40,41]; in addition, a sense of well-being and school adjustment develop over time and can predict each other [42]. Disadvantaged groups feel more discrimination, so they view their surroundings and the behavior of others more negatively, and become more emotionally sensitive; this, in turn, negatively predicts school adjustment [43]. Psychological adjustment can positively predict students' sense of school belonging. Accordingly, this study adopts psychological adjustment as the first mediating variable and school belonging as the second mediating variable, and assumes H4 as follows:

**H4.** *Psychological adjustment and school belonging play a chain mediating role between social support and professional identity (Figure 1).*

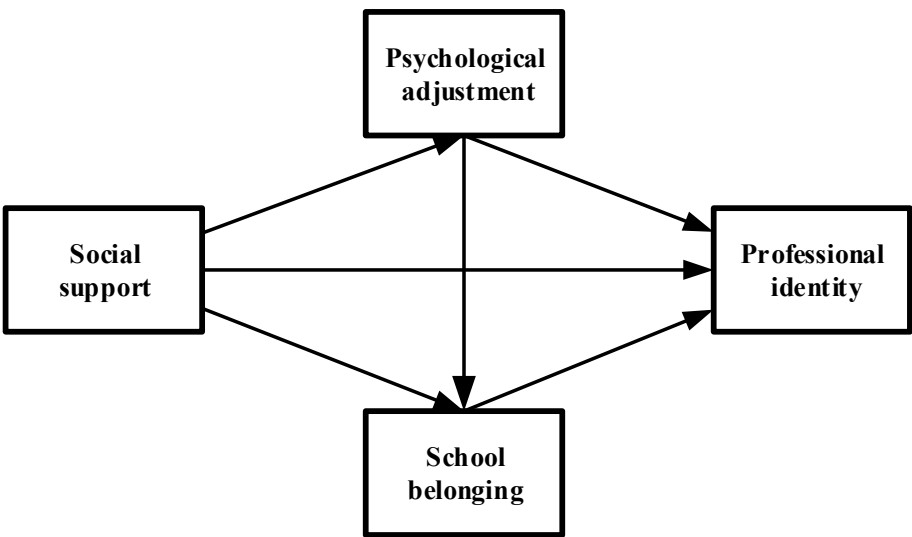

**Figure 1.** Research model.

## 2. Methods

### 2.1. Participants

A total of 377 third-year female students, majoring in the preschool nursery teacher program of six secondary vocational schools in Shanghai, participated in the current study. The data for the study were obtained from a project conducted to monitor and evaluate the quality of secondary vocational school preschool nursery programs in Shanghai, which

included third-year graduating students from all six secondary vocational schools in Shanghai that offer preschool nursery programs, all of whom were female; this may be due to the professional characteristics of preschool nursery programs, which result in an exclusively female student population. The Ethics Committee of Shanghai Normal University reviewed the research plan and approved this study. The official approval number is [2022] No. 084.

### 2.2. Measures

#### 2.2.1. Social Support

The students' perceived social support was assessed by the Child and Adolescent Social Support Scale (CASSS, 40 items) [44], which includes four dimensions: parental support (the perception of emotional, instrumental, informational, and appraisal support from parents, 10 items), teacher support (the perception of emotional, instrumental, informational, and appraisal support from teachers, 10 items), classmate support (the perception of emotional, instrumental, informational, and appraisal support from classmates, 10 items), and friend support (the perception of emotional, instrumental, informational, and appraisal support from friends, 10 items). For example, "parent makes suggestions when I am uncertain". A 6-point Likert scale from 1 (Never) to 6 (Always) was used. Higher scores indicate higher levels of social support perceived by the student. Cronbach's $\alpha$ of the four dimensions in the current study ranged from 0.91 to 0.96.

#### 2.2.2. Professional Identity

The professional identity was examined using the 23-item Professional Identity Questionnaire [7], which includes four dimensions: cognitive identity (degree of knowledge about the basic information of the major, 5 items), behavioral identity (students' learning behaviors regarding their major, 6 items), emotional identity (emotional preference for the major, 8 items) and professional appropriateness (the degree of matching between the major and oneself, 4 items). An example is, "I know the employment situation in my major". The items were assessed using a 5-point Likert scale from 1 (completely disagree) to 5 (completely agree). The Cronbach's $\alpha$ of the four dimensions ranged from 0.88 to 0.93 for the current study.

#### 2.2.3. Psychological Adjustment

Three indicators were used to investigate the psychological adjustment of the students, namely self-identity, well-being, and perceptions of discrimination. The Self-Identity Scale (SIS, 19 items), developed by Ochse and Plug in 1986, based on Erikson's theory, was used to examine students' self-identity [45]. An example is, "I feel certain about what I should do with my life". The items were assessed using a 4-point Likert scale from 1 (completely disagree) to 4 (completely agree). The Cronbach's $\alpha$ of the scale was 0.83 for this study.

A 28-item scale developed by the World Health Organization (WHO), called the Well-being Scale (WBS), was used to examine the students' subjective quality of life; in total, 5 of them (a shorter well-being index) were proposed by Bech and were used to measure the students' subjective well-being [46]. For example, "I feel calm" and "I feel energetic". The items were scored on a 6-point Likert scale from 1 (never) to 6 (always). The Cronbach's $\alpha$ of the scale was 0.74.

The Perception of Discrimination Scale was used to assess the students' perceived racial discrimination and interpersonal bias in their daily lives [47,48]. It contains one dimension with 13 items. In total, 10 items were used to assess the chronic, routine, and less overt experiences of discrimination that have occurred; then, 3 items were added to reflect perceptions of teacher discrimination. Sample items include the following: "Being treated with less courtesy than others" and "Your teachers treat you less with respect than other students". The items were scored on a 4-point Likert scale from 1 (never) to 4 (always). The Cronbach's $\alpha$ of the scale was 0.86.

### 2.2.4. School Belonging

An 18-item version of the Psychological Sense of School Membership Scale [31] was used to measure belonging or psychological school membership. The scale contained 1 dimension in order to measure students' school belonging in this study. Sample items include the following: "I feel like a real part of school", "Most of the teachers at school are interested in me" and "I am treated with as much respect as other students". All items use a 5-point Likert scale from 1 (completely disagree) to 5 (completely agree). The Cronbach's $\alpha$ of the scale was 0.91 for this study.

### 2.3. Data Analysis

SPSS (26.0) and AMOS (24.0) were employed for data analysis. Harman's single factor test was used to examine the common method bias for data validity. The descriptive data were reported using mean and standard deviation. Pearson correlation analysis was used to examine the correlation between each variable. Structural equation modeling (SEM) was performed using the maximum likelihood estimation method of AMOS, and the fit of the model was tested. Finally, bootstrapping ($n = 5000$ bootstrap samples) was carried out to investigate the direct effect between the social support and professional identity of the third-year students majoring in the preschool nursery teacher program of secondary vocational schools in Shanghai, and the mediating roles of psychological adjustment and school belonging.

## 3. Results

### 3.1. Common Method Bias

All measures in this study were self-reported scales filled in by students, which may lead to common method bias. Harman's single-factor method [49] was employed and the results found 22 common factors with eigenvalues greater than 1; the variance explained by the first factor was 29.46%, which was less than the critical criterion of 40%. Therefore, there was no significant common method bias in the data in this study.

### 3.2. Confirmatory Factor Analysis

The factor loadings of the three latent variables were between 0.520 and 0.938, the average variance extracted (AVE) was between 0.508 and 0.702, and the composite reliability (CR) was between 0.800–0.903. The results indicated that the measurements of the three latent variables have a good reliability and validity (Table 1).

**Table 1.** Results of Confirmatory factor analysis.

| Variable | Topic | Parameter Significance Estimation | | | | Factor Load | AVE | CR |
|---|---|---|---|---|---|---|---|---|
| | | Unstd. | S.E. | t-Value | *p* | Std. | | |
| SS | parental support | 1.000 | | | | 0.520 | 0.508 | 0.800 |
| | teacher support | 1.074 | 0.113 | 9.508 | *** | 0.645 | | |
| | classmate support | 1.419 | 0.153 | 9.297 | *** | 0.856 | | |
| | friend support | 1.132 | 0.121 | 9.320 | *** | 0.782 | | |
| PA | well-being | 1.000 | | | | 0.744 | 0.595 | 0.813 |
| | self-identity | 2.163 | 0.174 | 12.445 | *** | 0.887 | | |
| | perceptions of discrimination | −1.031 | 0.086 | −11.979 | *** | −0.666 | | |
| PI | cognitive | 1.000 | | | | 0.647 | 0.702 | 0.903 |
| | emotional | 3.018 | 0.216 | 13.954 | *** | 0.848 | | |
| | behavioral | 2.361 | 0.159 | 14.883 | *** | 0.938 | | |
| | appropriateness | 1.710 | 0.118 | 14.435 | *** | 0.889 | | |

**Note.** SS = Social support; PA = Psychological adjustment; PI = Professional identity; *** $p < 0.001$.

### 3.3. Descriptive Analysis and Correlation Analysis

The means, standard deviations, and correlation coefficients of the variables in the mediation model were reported in Table 2. The results of the correlation analysis indicated that social support was significantly positively correlated with professional identity (r = 0.59, $p < 0.01$), self-identity (r = 0.56, $p < 0.01$), well-being (r = 0.59, $p < 0.01$) and school belonging (r = 0.68, $p < 0.01$), but negatively correlated with discrimination perception (r = −0.50, $p < 0.01$). Moreover, students' s professional identity was significantly positively correlated with social support, self-identity (r = 0.53, $p < 0.01$), well-being (r = 0.50, $p < 0.01$) and school belonging (r = 0.67, $p < 0.01$), but negatively correlated with discrimination perception (r = −0.43, $p < 0.01$). Furthermore, school belonging was significantly positively correlated with self-identity (r = 0.67, $p < 0.01$) and well-being (r = 0.58, $p < 0.01$), but negatively correlated with discrimination perception (r = −0.52, $p < 0.01$).

**Table 2.** Mean, standard deviations, and correlations among variables (N = 377).

| Variable | *M* | *SD* | 1 | 2 | 3 | 4 | 5 |
|---|---|---|---|---|---|---|---|
| 1. social support | 199.02 | 26.40 | - | | | | |
| 2. professional identity | 95.93 | 14.81 | 0.59 ** | - | | | |
| 3. self-identity | 56.92 | 7.80 | 0.56 ** | 0.53 ** | - | | |
| 4. well-being | 21.63 | 4.29 | 0.59 ** | 0.50 ** | 0.66 ** | - | |
| 5. Perception of discrimination | 17.19 | 4.95 | −0.50 ** | −0.43 ** | −0.59 ** | −0.50 ** | - |
| 6. school belonging | 68.68 | 11.10 | 0.68 ** | 0.67 ** | 0.67 ** | 0.58 ** | −0.52 ** |

*Note.* ** $p < 0.01$.

### 3.4. A Chain Mediating Effect Analysis

To verify the mediating roles of psychological adjustment and school belonging between social support and professional identity, a fitting analysis of the conceptual framework mediation model was conducted using the AMOS 24.0 software package. The results of the model fit analysis were $\chi^2/df$ = 3.22, GFI = 0.94, CFI = 0.96, AGFI = 0.90, NFI = 0.95, IFI = 0.96, TLI = 0.95, RMSEA = 0.08, SRMR = 0.05. The data all met the criteria, indicating the good fit of the model [50].

The mediating bootstrap 95% CI effect was estimated using 5000 samplings, and the mediation effect test was carried out. If the bootstrap test showed that the CI did not contain a 0 value, it meant that the indirect effect was established. The test of the mediating model (Table 3) further showed that the total indirect effect of social support on the professional identity was significant, with total indirect effect = 0.33, 95% CI [0.05, 0.56], SE = 0.13, and that the ratio of the mediating effect to the total effect was 45.71%. The indirect effect contained one non-significant mediating pathway and two significant mediating pathways: Social support → psychological adjustment → professional identity, with indirect effect = 0.08, 95% CI [−0.11, 0.25], SE = 0.09. Social support → psychological adjustment → school belonging → professional identity, with indirect effect = 0.09, 95% CI [0.02, 0.18], SE = 0.04. Social support → school belonging → professional identity, with indirect effect = 0.16, 95% CI [0.07, 0.30], SE = 0.06. The contribution rates of the three indirect effects in the total effect were 10.55%, 12.38%, and 22.93%, respectively.

The standardized path coefficient model of social support, psychological adjustment and school belonging, affecting the professional identity, is shown in Figure 2. The path coefficient of social support → professional identity (β = 0.39, $p < 0.01$) was significant, indicating that social support has a direct effect on professional identity. The path coefficients of social support → school belonging (β = 0.52, $p < 0.001$) → professional identity (β = 0.31, $p < 0.001$) were significant, indicating that school belonging has a mediating effect between social support and professional identity. The path coefficients of social support → psychological adjustment (β = 0.79, $p < 0.001$) → professional identity (β = 0.09, $p > 0.05$) were not all significant, indicating that the mediating effect of psychological adjustment between social support and professional identity is not significant. The path coefficient of

social support → psychological adjustment → school belonging (β = 0.36, *p* < 0.001) → professional identity was significant, indicating that psychological adjustment and school belonging play an incomplete chain mediating role between social support and professional identity. Therefore, Hypotheses 1, 3, and 4 are supported.

**Table 3.** Mediating effect analysis.

| Pathway | Estimates | SE | Ratio | Bias-Corrected 95% CI | |
|---|---|---|---|---|---|
| | | | | *Lower* | *Upper* |
| SS → PA → PI | 0.075 | 0.093 | 10.55% | −0.112 | 0.249 |
| SS → PA → SB → PI | 0.088 | 0.039 | 12.38% | 0.020 | 0.178 |
| SS → SB → PI | 0.163 | 0.059 | 22.93% | 0.069 | 0.302 |
| Total indirect effect | 0.325 | 0.132 | 45.71% | 0.054 | 0.555 |
| Direct effect | 0.386 | 0.169 | 54.29% | 0.097 | 0.741 |
| Total effect | 0.711 | 0.050 | 100.00% | 0.605 | 0.802 |

*Note.* SS = Social support; PI = Professional identity; PA = Psychological adjustment; SB = School belonging.

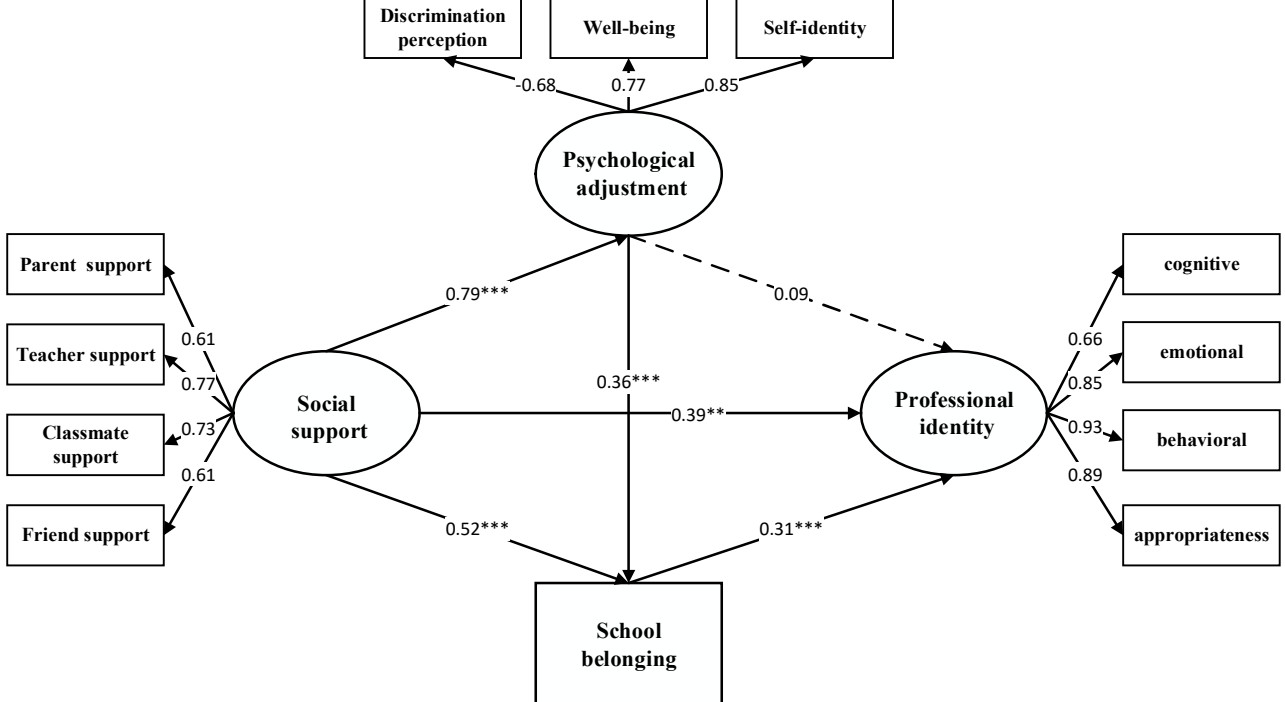

**Figure 2.** Intermediary Model of Association Between Social Support and Professional Identity. Note. *** *p* < 0.001, ** *p* < 0.01.

## 4. Discussion

As can be seen in the current study, social support not only directly influences the professional identity of students who major in the preschool nursery teacher program in secondary vocational schools, but also indirectly affects their professional identity using psychological adjustment and school belonging. That is, psychological adjustment and school belonging have a chain mediating effect between social support and professional identity. Support from parents, teachers, classmates and friends can promote students' psychological adjustment level, which can enhance students' school belonging and thus their professional identity.

### 4.1. The Relationship between Social Support and Professional Identity

Social support has a significant positive predictive effect on professional identity, which is consistent with existing research findings [18,19]. The contextualist developmental

view of sociocultural orientation emphasizes the process of identity integration, which is promoted by the interaction between individuals and social contexts [51,52]. Therefore, micro-environments, such as that created via interpersonal relationships between secondary vocational school students, as well as the macro-environment that is created via others' values and attitudes toward their professions, and the interaction between these two environments, affect students' emotional and behavioral integration into their own professions. Firstly, the support of parents and teachers gives students a deeper understanding of the professional situation, knowledge and skills. The parents in the family will provide students with more professional information, skills, and opportunities for professional exploration, enabling students to comprehend their majors from all aspects. Furthermore, the material and emotional support, and favorable family atmosphere, will contribute to students paying more attention to their professional learning. The professional and career guidance provided by teachers in schools accordingly improves students' knowledge of the profession, assisting them in perceiving and analyzing the values and attitudes of society as a macro-environment when approaching nursery-related careers. Positive feedback about the profession from parents and teachers leads students to develop positive emotions about the profession; meanwhile, students who have gained a lot of special knowledge believe that they are competent to perform tasks related to the profession and are thus willing to persist in the profession [53,54]. Secondly, support from adults and peers also brings positive emotional experiences and motivational beliefs [55], which alleviate the disapproval and academic stress of secondary vocational school students due to the poor employment situation of their majors; this positive emotion in their lives is transferred to their professional learning, thus leading to positive professional emotions among students. Finally, students in secondary vocational schools have low self-efficacy in learning their professions. Encouragement from adults and peers enhances students' self-efficacy, so that students try to persist in their professional tasks despite difficulties [56], consciously regulate their learning activities [57] and develop interest in their profession [58]. In summary, enhancing adult and peer support for students in all areas, especially in professional mentoring support and emotional support, facilitates the development of students' professional identity.

### 4.2. The Mediating Role of School Belonging

School belonging plays a partially mediating role between social support and professional identity, that is, social support leads secondary vocational school students to have a sense of school belonging, which in turn increases their level of professional identity. Moreover, they can gain a greater sense of security and the ability to explore the outside world from parental support early on, and social skills learned from parental and peer interactions can be applied to the school environment to develop satisfying school relationships, which can also lead to a sense of school belonging [41]. The support of teachers, peers, and friends provides a more inclusive microenvironment for students to feel satisfied and feel safe at school. In addition, high quality social network relationships provide a good social foundation for students to interact with teachers or peers to obtain more advice and school resources for professional learning [36,59], which in turn enhances students' professional cognition. Secondary vocational school students lack motivation to learn. Some studies suggest that an increased sense of belonging leads to increased motivation, engagement, and interest in academic learning [60]. The self-determination theory insists that when individuals receive external support to satisfy their basic psychological needs, such as a sense of autonomy and belonging, external motivation for learning will be transformed into internal motivation for students' active learning, thus taking the initiative to get rid of academic burnout, establishing professional learning goals, and taking the initiative to learn the profession. In this process, students' autonomous motivation and professional identity form a mutually reinforcing closed loop [61]. Both the belongingness hypothesis and Maslow's hierarchy of needs theory suggest that when individuals' belonging needs are not met, more negative emotions will be generated, leading to learning anxiety and problematic behaviors [62]. When students do not feel safe and satisfied in school, they also

do not have the means to invest more emotions and behaviors into their profession, lacking interest in professional learning, positive emotions, and the desire to explore, thus hindering their level of professional identity. Thus, school belonging improves the school life experience and thus affects students' professional identity by increasing their motivation to communicate and learn. Schools should encourage parents, teachers, and peers to provide support to students, provide a comfortable learning environment, and thus enhance students' sense of belonging in a school; this, in turn, increases their professional identity.

*4.3. The Chain-Mediating Role of Psychological Adjustment and School Belonging*

This study observes that social support affects the professional identity of secondary vocational students through a chain mediating effect of psychological adjustment and school belonging. That is, social support promotes secondary vocational students' psychological adjustment, and subsequently strengthens their sense of school belonging, thus facilitating the development of their professional identity. Secondary vocational school students are sensitive and lack self-confidence. Emotional and academic support from teachers can help students improve their status within the class, increase their sense of self-worth, reduce perceived discrimination within the class, and develop positive emotions. On the one hand, students are naturally more likely to build a sense of school belonging when they feel happy in school; when students perceive more discrimination, they develop negative attitudes towards their level of acceptance and develop negative emotions, and are unable to positively build school belonging [43]. On the other hand, according to the broaden-and-build theory, if students have a higher sense of well-being, they will have more flexible and abundant perceptions, thoughts, and behaviors, and will afterwards take this opportunity to experience school life positively [42]. Students who have a high level of school belonging are also more motivated and engaged in learning activities [63], which is behavior that reflects a higher level of professional identity and leads to higher professional satisfaction [64]. In addition, learning engagement also predicts higher professional identity [65]. with the support of parents, teachers and peers, it is easy to construct the self, accept the self, and can generate self-worth, thus students are more likely to accept their peers, teachers and establish good social connections at school, enhancing their school belonging [41,66]. Students with a sense of school belonging are more satisfied with their school and this satisfaction may be transferred to the profession; then, there are higher expectations in the profession, thus enhancing the professional identity of students [36]. Therefore, high levels of self-identity and well-being, as well as low levels of discrimination, are conducive to adolescents ability to build a sense of belonging in school; this, in turn, is more conducive to generating the internal motivation to learn and engage in professional learning, to generating higher professional expectations, and thus enhancing their professional identity. In conclusion, in order to enhance students' professional identity, social support should also improve students' psychological adjustment and school belonging.

Notably, this study illustrates that the intermediary role of psychological adjustment between social support and professional identity is not significant. In contrast, the mediating path of "social support → school belonging → professional identity" and "social support → psychological adjustment → school belonging → professional identity" has more significance in terms of the mechanism of social support affecting professional identity. It may result from the fact that school is the first formal exposure to the profession; this is more closely related to the profession compared to the general psychological adjustment resources that are available in other domains of daily life, so the development of professional identity may be more strongly influenced by proximal environmental variables. Professional identity is not constructed by the individual adolescent alone, but is more likely based on interactions between individuals and significant others [67]. This explanation can also be corroborated by the effect of the group on Chinese collectivist cultural values. School belonging better reflects the quality of the connections that individuals make with teachers and peers in the schools. Moreover, school belonging and professional identity embody a degree of personal identification or school belonging in the environment

and profession, respectively, both of which are closer from a conceptual perspective [36]. This leads to the failure of the mediating role of psychological adjustment, meaning that psychological adjustment must be applied to the proximal environment (school); this is more relevant to professional identity, and professional identity that comes from significant and satisfying school experiences and interpersonal interactions.

## 5. Conclusions, Limitations, and Implications

The present study explores the mediating role of psychological adjustment and school belonging between social support and professional identity, and finds that social support has a significant positive predictive effect on secondary vocational school students' professional identity, and that psychological adjustment and school belonging play a partial mediating role between social support and professional identity. This also shows that the formation of secondary vocational school students' professional identity is the outcome of both personal and background environments. For the purpose of improving students' professional identity, support must be given to peers in order to establish a favorable school atmosphere.

There are many limitations to this study. First, all of the participants in this study were female students, and whether the male population would also satisfy the set of hypothesized relationships deserves further exploration. Therefore, future research is needed to expand the sample size and balance the gender ratio. Second, the data adopted in this study are cross-sectional rather than longitudinal, and future research should consider the variation in the relationships between variables over time. Consequently, although this study verifies the relationship between social support and professional identity, as well as the chain mediating mechanism of psychological adjustment and school belonging, future studies are still needed to include more important variables to enrich the relationship mechanism between them.

The results of this study are expected to give us a methodology that can be used to improve the professional identity of secondary vocational school students; others should be encouraged to provide more social support to students, especially the support of significant others in the school. This support should be aimed at improving students' psychological health, such as alleviating their negative emotions and recognizing their self-worth. Schools should also build a good school network around students to foster school belonging and improve students' school satisfaction, thus enhancing students' professional identity in both direct and indirect ways. This study verifies the positive effect of social support on the professional identity of secondary vocational school students' professional identity and reveals the mechanism chain between psychological adjustment and school belonging, which provides a new path and perspective for the construction model of the professional identity of secondary vocational school students. In addition, this study also uncovers the two-way relationship between school belonging and psychological adjustment by enriching the role of school belonging, providing a theoretical basis and practical ideas for the construction of a "soft environment" for talent cultivation in vocational schools.

**Author Contributions:** Conceptualization, H.H.; Funding acquisition, H.H.; Investigation, Y.C.; Methodology, Y.C.; Supervision, H.H.; Writing–original draft, Y.C. and H.H.; Writing—review and editing, Y.Y. and H.H. All authors have read and agreed to the published version of the manuscript.

**Funding:** This research received no external funding.

**Institutional Review Board Statement:** The study was conducted in accordance with the Declaration of Helsinki and approved by the Ethics Committee of Shanghai Normal University (No. 084-2022).

**Informed Consent Statement:** Informed consent was obtained from all subjects involved in the study.

**Data Availability Statement:** The data presented in this study are available on request from the corresponding author. The data are not publicly available due to ethical requirements.

**Conflicts of Interest:** The authors declare no conflict of interest.

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
