# Peer review of "Effects of Social Support on Professional Identity of Secondary Vocational Students Major in Preschool Nursery Teacher Program: A Chain Mediating Model of Psychological Adjustment and School Belonging"

_sustainability, doi:10.3390/su15065134_

Round 1

Reviewer 1 Report

The article is informative and comprehensive.

The abstract is clear and includes all expected elements.

The Ethics approval usually come with a number, could this be included? 

Suggestion: what of student's self-efficacy, self-confidence to complete the nursery job?

Item 3.3 Line 244 needs a reference to support observation.

The authors could consider presenting a factor analysis, how do the items load?

Spelling: H3 (Line 125) "an" should be "a".

2.3 (Line 208) "data of were" - data of what? were reported?

Reviewer 2 Report

Summary 

The aim of this paper is to contribute to the knowledge of the effects of social support on professional identity using a Chain Mediating Model of Psychological Adjustment and School Belonging, based on a survey investigating 337 third-year students from Secondary Vocational Students Major in Preschool Nursery Teacher Program in China

The result is of high relevance for the research field and its findings, that social support, professional identity, school belonging, and psychological adjustment (well-being, self-identity) correlated with each other in a significant and positive way may also be of great importance for the development of the Secondary Vocational Students Major in Preschool Nursery Teacher Programs in China. However, it may also contribute to the development of preschool teacher programs in other parts of the world.

General concept comments

Article: I perceive the article's main contribution to be its complex and innovative methodology and extensive amount of data. At the same time, the many different variables in the Chain Mediating Model that are examined can appear as a weakness, as it is somewhat difficult to follow the reasoning and to clearly perceive the essentials of the result.

Perhaps the authors can clarify in particular what "psychological adjustment" can be about when it is introduced. English is not my mother tongue, but I am not sure if the use of "psychological" is the most fitting word. In the results, the relationship between the hypotheses that are set and the different parts of the results can also be made more explicit.

Review: Secondary vocational school students have, as being adolescents, according to the authors a specific need for multidimensional social support to reinforce their self-awareness and employment preparation. The mechanism underlying the relationship between social support and professional identity is pointed out as unclear and as such, something in need of investigation and as a gap in knowledge. Thus there is a clear motivation for the contribution of the findings in the article.

I find that the topic is of high relevance for research that has to do with childcare students' development of professional identity and sense of school belonging. The covered subject has been described and treated thoroughly and essentially clarifyingly. The references are used to substantiate the content in a relevant way. However, a reference to sources on the research mentioned in the introduction should be entered in the text (line 38). An intratextual reference could, for example, be sufficient.

Specific comments 

Line

Comment

38

A reference is needed after the sentence: “Nonetheless, previous research has indicated that social status, income, and the work intensity of preschool nursery teachers are underestimated”.

81

program should be programs (plural).

82

Punctuation before the heading should be removed

84

Unclear sentence: psychology or psychological condition?

102

an mediating – should be a mediating

119

had been – should be have been

125

an mediating – should be a mediating

135

Remove the word being after well-being and change and to as (well as)

166

Remove is

185

Change were to was

208

Remove of

243-244

Suggestion: indicating that the model was appropriate

353

Remove on

354

Change be resulted to result

392-395

The builders of Chinese-style modernization ought to identify with their professions and love their occupational positions, to continuously improve their vocational skills and make unique contributions.

This is a very normative way to write. Even though this is part of the conclusion a more modest and reflexive expression would be more appropriate.

Reviewer 3 Report

Thank you for allowing me to review the manuscript entitled 'Effects of Social Support on Professional Identity of Secondary Vocational Students Major in Preschool Nursery Teacher Program: A Chain Mediating Model of Psychological Adjustment and School Belonging'.

The article is interesting and timely. It addresses a topic very dear to my heart and I congratulate the authors for the valuable insights from their findings. 

The introduction well presents the topic of the manuscript and the literature review is up-to-date. Below are some suggestions for improving the quality of the paper that I hope will be helpful to the authors:

- In the introduction, the authors refer to "professional identity" and not "vocational identity." I would ask the authors to clarify then which current they refer to in the study of identity.

- Why do the authors have a sample of all females? I would ask the authors to clarify whether this was an a-priori choice of the study or not. In either case it should be justified.

- I would ask the authors to go into the specifics of the study context in the introduction. It will also be important in discussing the results.

- Why 2000 bootstrap samples and not 5000?

- Are the reported betas standardized?

- In the discussions I would ask the authors to discuss the results in light of the literature on the topic and the specific study context. Just latching onto this, it would be desirable for the authors stress the possible practical implications of their study. 

Round 2

Reviewer 3 Report

Thank you for revising the paper in the light of my comments/suggestions. in my opinion, the paper can be accepted in its current form.